# The Molecular ‘Myc-anisms’ Behind Myc-Driven Tumorigenesis and the Relevant Myc-Directed Therapeutics

**DOI:** 10.3390/ijms21249486

**Published:** 2020-12-13

**Authors:** Jessica McAnulty, Analisa DiFeo

**Affiliations:** Department of Pathology, University of Michigan, Ann Arbor, MI 48109, USA; jlmc@umich.edu

**Keywords:** myc, cancer, inhibitors, transcription, stability, max, cell cycle, metabolism, synthetic lethality

## Abstract

*MYC*, a well-studied proto-oncogene that is overexpressed in >20% of tumors across all cancers, is classically known as “undruggable” due to its crucial roles in cell processes and its lack of a drug binding pocket. Four decades of research and creativity led to the discovery of a myriad of indirect (and now some direct!) therapeutic strategies targeting Myc. This review explores the various mechanisms in which Myc promotes cancer and highlights five key therapeutic approaches to disrupt Myc, including transcription, Myc-Max dimerization, protein stability, cell cycle regulation, and metabolism, in order to develop more specific Myc-directed therapies.

## 1. Myc’s Role as a Transcription Factor

Think about any key cellular process, and the Myc family most likely has a role in it: proliferation, metabolism, differentiation, and apoptosis. The Myc transcription factor family consists of c-Myc, N-Myc, and L-Myc. Discovery of c-Myc led to finding N-Myc, primarily expressed during development or in neuroblastoma, and L-Myc, expressed in lung tissue and small-cell lung cancer [1,2]. Although the family shares stretches of homologous regions and some related targets, N-Myc and L-Myc are less characterized. c-Myc (hereon referred to as Myc), is the most well-studied family member as it is an influential protooncogenic transcription factor that binds to about 15% of genes [3,4,5]. In order to regulate gene expression, Myc recruits or interacts with many different cofactors, including histone acetyl transferases (CBP, p300, GCN5/TRAPP), P-TEFb and polymerases, and chromatin remodelers (BRD4, the SWI/SNF complex, SIRT1) [5,6]. It is important to note that Myc can also repress gene expression by binding to the promoter region and interacting with MIZ1 and SP1 to displace co-activators or by recruiting DNA methyltransferases [5]. Furthermore, there are two structural components of the *MYC* gene that are essential to drive its role as a transcription factor: the E box and the basic helix-loop-helix leucine zipper domain.

The canonical Myc E Box DNA binding motif (5′-CACGTG-3′) is one of the most frequent regulatory motifs in the human genome [7]. Although Myc is not the only transcription factor that can occupy this motif, elevated levels of Myc will replace the other bound transcription factors [8], demonstrating how Myc can influence the transcription of many genes and diverse processes in proliferating cells. A Myc core signature of 50 common target genes across four human cancer cell types and human embryonic stem cells revealed Myc’s influence in RNA processing and ribosome biogenesis [9]. Other diverse functions of Myc target genes include cell cycle regulation, metabolism, cell adhesion, and signal transduction [5,10,11].

However, Myc does not exclusively bind to the E-box to modulate transcription. In repression of gene transcription, cofactors recruit Myc to the promoters lacking the E-box and interfere with active transcription factors [12,13,14]. Furthermore, Myc can amplify transcriptional signals by accumulating at the promoters of active genes, even in those with low-affinity E-box-like sequences [15,16]. There is still debate of whether Myc drives global amplification of transcription [15,16] or if global amplification is an indirect consequence of Myc’s selective regulation of gene targets [17,18,19].

In addition to the E box binding motif, the basic helix-loop-helix leucine zipper (bHLHZip) domain is crucial for Myc’s activity. To take on its role as a transcription factor, Myc must heterodimerize with Myc-associated factor X (Max); Myc is incapable of homodimerizing and is inactive as a monomer. Max binds to Myc at the bHLHZip domain [20,21], and this heterodimerization is required to bind to the E box consensus sequence and activate transcription [22,23]. However, overexpression of Max leads to transcriptional repression as the Max homodimers antagonize Myc/Max heterodimers [22,24]. Mad, a transcriptional repressor, can also reduce Myc-driven transcription by dimerizing with Max [5].

## 2. Dysregulation of Myc Leads to Cancer

Normally, Myc expression is tightly controlled at each molecular level (transcriptionally, post-transcriptionally, translationally, and post-translationally via protein stability, and via protein interactions), and has a short half-life of 20–30 min [25,26,27,28,29]. Given that there are many levels of regulation, as a consequence, there are many opportunities for which control of *MYC* can go awry. For instance, point mutations, chromosomal translocations, and gene amplification, or other factors that activate transcription or stabilize Myc, have been found in a wide range of cancers, which are further described by Meyer and Penn and Kalkat et al. [30,31]. This oncogenic activation, which leads to sustained levels of Myc, contributes to tumorigenesis and evasion of tumor-suppressive checkpoints leading to uncontrolled cell growth. *MYC* expressing tumors thus become addicted to and depend on the oncogene, as shown in cancer models with conditional activation of *MYC* [32]. On the contrary, inactivation of *MYC* leads to tumor regression in transgenic mouse models, displaying Myc’s vital role in tumor initiation and maintenance [33,34,35].

*MYC* amplification is found in 21% of patients across 33 different cancers [36], particularly breast cancer, lung squamous cell carcinoma, uterine carcinoma, esophageal carcinoma, and ovarian cancer [25] (Figure 1). The highest rates of amplification are seen in high-grade serous ovarian cancer wherein greater than 50% of tumors harbor this genomic alteration. *MYC* translocation affects several hematological malignancies, including multiple myeloma, Burkitt’s lymphoma, diffuse large cell lymphoma, and T-cell acute leukemia [37]. Alternatively, some tumors that do not display *MYC* amplification show extreme phosphorylation levels which aid in Myc stability [38,39,40,41].

With Myc’s prominent role across many cancers, the idea of Myc as a clinical target is too good to be true. Although targeted inhibition of *MYC* via siRNA reduces tumor burden in mice with very few toxicities despite Myc’s influence on global transcription [33,35,42,43], global *MYC* knockout is embryonic lethal in mice. Thus, cautious measures in observing side effects of disrupting Myc need to be addressed [44]. The less expected problem is that direct inhibition of Myc is not possible with current therapeutic approaches—Myc lacks both enzymatic activity and an active site for a small molecule to disrupt protein-protein interactions [45]. Myc’s primary nuclear localization further escalates the problem. Nonetheless, scientific discoveries led to creative ways to downregulate Myc. This review focuses on how Myc’s oncogenic activation leads to tumorigenesis through initiating transcription, increasing stability, and influencing cell cycle and metabolism, coupled with descriptions of the indirect inhibitors of Myc that target each mechanism (Figure 2). The molecular changes in which *MYC* becomes an oncogene (mutations, translocations, and amplification) is beyond the scope of this review [30,31,32].

## 3. Disrupting Myc Stability to Inhibit Its Actions as a Transcription Factor

In cancer, Myc’s aberrant function as a transcription factor leads to increased cell proliferation, cell differentiation, cell adhesion, and angiogenesis [10]. Here we will focus on inhibiting transcription, disrupting Myc/Max dimerization, and enhancing protein degradation as strategies to disrupt Myc gene and protein stability and therefore Myc-driven tumorigenesis.

### 3.1. Myc Drives Aberrant Transcription

As discussed, *MYC* amplification is common among many cancer types. This amplification of *MYC* results in increased binding of Myc to promoters and enhancers of active genes, which magnifies the transcriptional signal [15,16] and as a consequence, increases global transcription. During transcription, Myc recruits the transcriptional pause-release complex P-TEFb (a heterodimer of cyclin-dependent kinase 9 (CDK9) and cyclin T1, T2, or K) [6,16,46]. P-TEFb leads to activation of transcriptional elongation by phosphorylating RNA polymerase II (Pol II) via CDK9, stimulating pause release [47,48,49]. Furthermore, BRD4, part of the bromodomain and extra-terminal motif (BET) protein family, also recruits P-TEFb to promoters to initiate transcription elongation [50]. The overlapping roles of BET proteins and Myc in recruiting P-TEFb suggests BET proteins or CDK9 as therapeutic targets. First, BET proteins are known to regulate *MYC* transcription [51]. A recent study demonstrated in normal cells that BRD4 has even more control over Myc by binding and phosphorylating Threonine 58 on Myc, leading to degradation [52]. However, Myc is also capable of regulating BRD4′s histone acetyl transferase activity [52]. Additional studies are needed to better understand how this circular balance may be affected in cancer. CDK9 is a potential target as it is part of P-TEFb, is necessary for proliferation and maintenance of *MYC*-overexpressing hepatocellular carcinoma [53], and is required for maintenance of gene silencing in several cancer cell lines [54]. Another tumorigenic feature of Myc is looping to tumor-specific super-enhancers (sites defined by multiple enhancers abnormally bound by a plethora of transcription factors, such as BRD4 and CDK9) [55]. Therefore, inhibiting *MYC* transcription indirectly via BET inhibitors or affecting transcription of Myc target genes by inhibiting CDK9 are promising strategies that have shown efficacy in Myc-driven cancers (Figure 3).

#### Targeting MYC Transcription—BET Inhibitors, BRD4 Degraders, CDK9 Inhibitors

The BET proteins, BRD2, BRD3, BRD4, and testis-specific BRDT, are epigenetic readers and histone acetyl transferases that activate transcription via binding to specific acetylated lysine residues on histone tails. The bound BET proteins regulate chromatin remodeling via H3K122 acetylation and act as scaffolds to form transcription complexes by recruiting transcriptional activators such as P-TEFb [50,56,57]. Furthermore, BRD4 influences mitotic progression by binding selectively to transcriptional start sites of M/G1 genes [58]. Oncogenes, such as *MYC*, have a transcriptional dependency on BRD4 and recent findings suggest additional non-transcriptional functions of BRD4 in cancer [59]. Bromodomains have a mostly hydrophobic pocket with aromatic rings and is an ideal size for protein–protein interactions, making bromodomains attractive and obtainable therapeutic targets, unlike Myc [56]. BET protein inhibitors compete for access to the bromodomain and upon binding, disrupt chromatin remodeling and prohibit expression of target genes, including *MYC*. Filippakopoulos et al. and Nicodeme et al. independently designed some of the first bromodomain inhibitors, known as JQ1 and iBET respectively, that are highly specific towards the BET protein family [60,61]. Initial studies showed efficacy of JQ1 downregulating both *MYC* expression and Myc’s transcriptome genome-wide in Myc-addicted hematological malignancies [62,63,64], and solid cancers [65,66,67,68]. iBET’s proof-of-concept in preventing BET proteins from binding to acetylated histones was demonstrated in an inflammation context [61], although a follow-up study exhibited that iBET was capable of downregulating *MYC* expression, but to a lesser extent than JQ1 [51]. It is important to note that both JQ1 and iBET lack specificity for a particular BET protein family member, which limits their therapeutic availability [69,70,71]. Therefore, these BET inhibitors serve best as tools to improve our understanding of targeting bromodomains and the effects on *MYC*. The discovery of JQ1 and iBET inspired the development of additional BET inhibitors, with 10 inhibitors being assessed in clinical trials, including MK-8628/OTX015. Phase Ib trials included six solid tumors such as NUT midline carcinoma (NMC), which harbors an oncogenic form of *BRD4*, known as *BRD4-NUT*. The trial (NCT02259114) completed with a recommended dose for Phase II studies, although the NMC patients that initially responded, relapsed several months after treatment [72]. BET inhibitors as a whole currently appear to have limited therapeutic response and dose-limiting toxicities. More preclinical research will increase the biological knowledge on mechanisms of action and resistance of BET inhibitors.

In addition to BET inhibitors, there are also BET degraders that utilize a concept designed in 2000: PROteolysis TArgeting Chimeric (PROTAC) [73]. PROTAC protein degraders link the protein of interest to an E3 ligase in order to ubiquitinate the protein of interest for degradation. This approach has been adapted to a variety of targets, including the androgen receptor, estrogen receptor, BCL2, CDK9, and BET proteins to name a few [74]. PROTAC technology has entered clinical trials, including Arvinas’s ARV-110 for patients with metastatic castration-resistant prostate cancer (NCT03888612) which has shown efficacy and a promising safety profile in Phase I [75]. The first PROTAC BET degraders, including MZ1, a BRD4-specific degrader, were designed in 2015 and demonstrated increased apoptotic response compared to nonspecific BET inhibitors, but a modest decrease in *MYC* expression [76,77].

It appears the antitumor efficacy of both BET inhibitors and BET degraders is most likely due to global transcription downregulation, rather than downregulation in *MYC* transcripts specifically [78]. Devaiah et al. recently discovered crucial molecular differences in Myc stability between BET inhibitors and BET degraders. Since endogenous BRD4 destabilizes Myc, treatment with a BRD4 degrader, such as MZ1, enhances Myc stability, but treatment with a BET inhibitor, such as JQ1, does not affect BRD4′s phosphorylation of Myc and therefore Myc’s half-life is unaffected while *MYC* transcription is downregulated [52]. Several PROTAC BRD4 degraders demonstrate robust decreases of *MYC* expression throughout 3–24 h [79,80], though there are no current clinical trials on BET/BRD4 degraders. Perhaps later timepoints and investigation of phophorylated-S62-c-Myc expression will aid in understanding long-term effects of BET degraders on Myc stability. For further reading, detailed reviews on bromodomains and their inhibitors are cited [56,81,82,83,84].

CDK9 is another potential therapeutic target, given its kinase activity in the P-TEFb complex which releases paused RNA Pol II to initiate transcription. CDK9 inhibitors demonstrate efficacy in downregulating *MYC* transcripts and Myc stability across hepatocellular carcinoma [53], mixed-lineage leukemia [85], diffuse large B-cell lymphoma [86], acute myeloid leukemia [87], and pancreatic cancer [88]. Although preclinical studies have shown efficacy in targeting CDK9, the sequence similarity to other cyclin-dependent kinases made specificity difficult. However, several groups succeeded in creating CDK9-specific inhibitors and PROTAC degraders [38,54,89]. A recent study demonstrated that CDK9-specific inhibitor, MC180295, downregulates *MYC* and leads to reactivation of epigenetically silenced tumor suppressor genes [54]. Thus, downregulation of *MYC* is not due to off-target effects of nonspecific CDK inhibition. Initial nonselective CDK inhibitors did not succeed in clinical trials, most likely due to toxicities from off-target effects. These trials included patients of many cancer types and were not selective to *MYC*-amplified patients [90]. However, CDK9-specific inhibitors, such as BAY 1143572 (NCT01938638), are beginning to enter clinical trials in patients with advanced cancer and will evaluate *MYC* expression as a biomarker [91,92].

Additionally, combining CDK9 and BET inhibitors synergistically improves anti-proliferative activity in several cancers, with no hematological toxicity or weight loss shown in vivo [93,94,95]. Of the same note, BET inhibitors were also efficacious when paired with additional inhibitors, such as PI3K, ERK, or BCL2 inhibitors [81]. Readers are referred to reviews further discussing targetable Myc cofactors that aid in tumorigenesis [13], such as G quadraplex stabilizers [96,97]. In all, BET and CDK9 inhibitors vastly affect transcription and as a result, downregulate *MYC* expression indirectly; improving their specificity is expected to increase their therapeutic benefit.

### 3.2. Myc/Max Dimerization

Another way of affecting Myc transcriptional stability is by preventing Myc from interacting with DNA. Myc must dimerize with Max in order to drive gene expression, though a recent structural study demonstrates that against previous belief, Myc is stabile in the absence of binding DNA [45]. Although, Max heterodimerization with Myc is required for Myc’s oncogenic activity [98]. Therefore, inhibiting Myc and Max dimerization prevents Myc from initiating gene transcription. There are two immediate challenges: (a) targeting the bHLHZip domain is nonspecific to Myc/Max and therefore could present off-target effects and (b) there are no apparent pockets for which a small molecule can bind [99,100]. Despite this, there has been success in disrupting the Myc/Max interaction with several mini-proteins or molecules, including Omomyc, 10058-FA, 10074-G5, KJ-Pyr-9, MYCMI-6, and KI-MS2-008 (Figure 4).

#### Disrupting Myc/Max Dimerization

The most well-known, and perhaps the first, Myc/Max dimerization inhibitor is Omomyc, a dominant negative mutant of Myc’s bHLHZip domain with 4 amino acid mutations in the leucine zipper that prevents Myc/Max heterodimerization [101]. Omomyc was a laboratory tool developed to bind and inhibit Myc. Over the past two decades, research produced a better understanding of how the molecular tool functions: Omomyc reduces the amount of Myc that can bind to promoters by either heterodimerizing with Myc in the cytoplasm, heterodimerizing with Max, or homodimerizing. Recent data show Omomyc preferentially binds to Max or homodimerizes [102]. The Omomyc homodimers or Max/Omomyc heterodimers are transcriptionally inactive complexes that bind specifically to E-box sequences and displace Myc/Max heterodimers resulting in decreased Myc-driven transcription [102,103,104,105]. Importantly, Omomyc is specific towards Myc’s function and does not suppress gene expression of other E-box-binding transcription factors [103].

Omomyc has shown efficacy in several tumor studies when it is conditionally or transiently expressed in the cell or linked with a cell penetrating Phylomer [43,106,107,108]. However, Omomyc is indeed capable of penetrating cells, including non-small cell lung cancer, neuroblastoma, glioblastoma, and melanoma cell lines, due to its basic region [103]. Until recently, in vivo proof-of-concept was lacking. Beaulieu et al. show Omomyc downregulates Myc target gene expression and prevents tumor progression in lung adenocarcinoma in vivo models via intranasal administration (2.37 mg/kg) over four weeks [103]. Similarly, in a lung adenocarcinoma xenograft model, paclitaxel combined with Omomyc administered intravenously diminished tumor growth over 30 days [103]. Both models showed no significant changes nor toxicities in blood counts or pathology reports of all major organs. In non-tumor-bearing mice, Demma et al. show Omomyc injected intravenously (5.22 mg/kg) primarily distributes to the liver and kidneys and has a short half-life in plasma [102]. Although this study used a higher dosage of Omomyc than the cancer study, toxicities of Omomyc in normal cells must be considered in future preclinical studies. Dr. Soucek, who created and studied Omomyc over the past 20 years, created the company Peptomyc to develop and sponsor Omomyc-derived clinical candidates; the first clinical trial is anticipated to start in 2021.

While Omomyc is capable of disrupting Myc/Max dimerization *and* preventing Myc from interacting with DNA, other Myc/Max inhibitors are typically characterized by one of those two actions. In 2002, Berg et al. demonstrated the proof-of-concept of using combinatorial chemical libraries to find small molecule inhibitors of protein–protein interactions, including Myc/Max [109]. Shortly after, Yin et al. identified specific Myc/Max inhibitors from a combinatorial library including 10058-FA and 10074-G5, that result in G0/G1 cell cycle arrest and apoptosis in vitro [110]. However, in vivo studies show both 10058-FA and 10074-G5 are rapidly metabolized and lack anti-tumor activity [111,112]. Therefore, these compounds best serve as molecular tools and a starting point for new compound development. More recent Myc/Max inhibitors include KJ-Pyr-9 and MYCMI-6. KJ-Pyr-9 has sufficient pharmacokinetic properties to penetrate tissue and prevent tumor growth, but cannot reduce existing tumors; its inability to decrease tumor size may be due to residual Myc activity as an effect of incomplete Myc inhibition [113]. MYCMI-6 was identified in 2018 as a promising Myc/Max-specific inhibitor that halts Myc-driven transcription, induces apoptosis, and reduced tumor proliferation in vivo [114]. Interestingly, these different Myc/Max inhibitors all initiate different biological effects.

Recently, Han et al. identified novel Myc-binding inhibitors, MYCi361 and MYCi975, that appear to act through disrupting Myc/Max dimers and increasing Threonine (T)58 phosphorylation of Myc, which leads to Myc degradation. MYCi-induced degradation could be a result of changes in Myc confirmation as it interacts with MYCi; it is important to note that not all Myc/Max inhibitors lead to Myc degradation [115]. Treating cells with proteasome inhibitor MG132 or exposing non-phosphorylatable Myc (T58A mutant) cells to MYCi361 rescues or prevents the MYCi361-induced Myc degradation [116]. In vivo studies utilized a Myc-driven prostate cancer mouse model, MycCaP, in which tumors were significantly decreased upon MYCi361 treatment. Additional studies are required to determine its efficacy in other cancers. Given that MYCi treatment modified the tumor microenvironment through increased expression of PD-L1, Han et al. demonstrated synergistic effects with MYCi361 and anti-PD1 in the MycCaP model, despite the model’s documented resistance to anti-PD1 therapy [116]. MYCi975 performs similarly to MYCi361 but has a higher therapeutic index and better tolerability in vivo of up to ten time the anti-tumor efficacious dose. There is promise for future studies on MYCi975 due to its inhibition of cancer cell growth and reduction of Myc target gene expression in vitro and decreased tumor growth in vivo with high tolerability. These compounds represent a new class of directly targeting Myc and inhibiting Myc/Max dimers, which led to Myc degradation.

Agents that inhibit Myc/Max from binding DNA have also been pursued, although they lack in vivo data and specificity towards Myc/Max [117]. One approach to prevent Myc binding to DNA is by targeting one of the many cofactors that recruits Myc to its target genes. WDR5 is an adapter protein that interacts with histone methyltransferase and serves as a scaffold for chromatin; it recruits Myc to chromatin and the Myc-WDR5 interaction is required for Myc-driven tumorigenesis [118]. Thomas et al. recently discovered that WDR5 stabilizes the Myc/Max interaction with DNA and a mutant Myc that cannot bind to WDR5 leads to tumor regression in a Burkitt lymphoma in vivo model [118]. However, the mutant Myc was capable of binding to chromatin, suggesting that targeting WDR5 does not affect Myc’s ability to interact with DNA. Given the antitumor effect and the druggable pockets within WDR5, it is a viable anti-Myc contender to pursue; additional recent advances with WDR5 are described in the Metabolism section of this review.

Alternative approaches that are not widely explored include stabilizing Max. In 2019, Struntz et al. discovered KI-MS2-008, which stabilizes Max homodimers while decreasing both Myc binding at promoters and Myc protein levels [119]. KI-MS2-008 proved efficacious in T cell acute lymphoblastic leukemia and hepatocellular carcinoma in vivo models with a reduction in tumor burden and no toxicities in liver or kidney [119]. Further studies are needed to determine the mechanism of action to optimize for in vivo use, but for now, KI-MS2-008 serves as an instrument to investigate the importance of Max dimerization in cancer.

Exploring the Myc/Max interaction has been a popular avenue for disrupting Myc-driven transcription. Sammak et al.’s high resolution crystal structure of the Myc and Max heterodimer in the absence of DNA will aid in development of future Myc-targeting therapeutics [45]. Pursing additional compound libraries, such as Carabet and colleagues’ computational screen to discover inhibitors of Myc-max in silico, can further broaden our understanding of inhibiting Myc/Max dimers [99]. Future Myc/Max dimerization inhibitors must overcome challenges faced by current therapeutics such as fast metabolism, poor penetrability, and nonspecific targets.

### 3.3. Myc Protein Stability

The short half-life of Myc is evidence for Myc’s highly controlled turnover. Myc’s stability is regulated by phosphorylation on serine 62 (S62) and threonine 58 (T58) by several proteins through the Raf-MEK-ERK kinase and phosphatidylinositol-3 kinase (PI3K)-Akt pathways [28,120]. First, extracellular signal-regulated kinase (ERK), CDK1, or growth signals stabilize Myc by phosphorylating S62. Glycogen synthase kinase 3 (GSK3) is recruited to phosphorylate T58, which is required for Myc degradation. In brief, Pin1 isomerizes proline 63 on Myc, in which protein phosphatase 2A (PP2A), a serine/threonine phosphatase, can now dephosphorylate Myc at S62 [121]. The unstable Myc, with only T58 phosphorylation remaining, becomes ubiquitinated by Fbw7 and is sent for degradation [122]. Again, these many levels of regulation provide multiple opportunities for cancer hijacking. In cancers that lack *MYC* amplification, there are increases in the stabilizing pS62-Myc and decreases in the degrading pT58-Myc, therefore promoting Myc’s stability and activity [38,39,40,41]. Studies show that mutating T58 to alanine, a non-phosphorylatable residue, results in stable Myc expression and tumorigenic properties, suggesting Myc stability has a role in transformation [123,124,125].

In cancer, we see faulty regulation of these proteins that modify the phosphorylation on Myc and promote stabilization. We will describe three scenarios—activation of PI3K/AKT signaling (which inhibits GSK3B), overexpression of Pin1, and suppression of PP2A activity—that stabilize Myc. PI3K, PTEN, and upstream components of the PI3K/AKT pathway are commonly mutated in cancer to promote pathway activation [126]. Activated AKT phosphorylates (and therefore inhibits) GSK3, which in turn enhances Myc stability [127,128] as GSK3 cannot phosphorylate T58-Myc. This is just one example of how an upstream signaling pathway (MAPK, Wnt, Notch) can quickly trickle down to promoting cancer through Myc.

Peptidyl-prolyl cis-trans isomerase NIMA-interacting 1 (Pin1), an isomerase that specifically recognizes the serine/threonine-proline motif, is overexpressed in several cancers including pancreatic, breast, and prostate and its expression correlates with poor clinical outcomes [129,130,131]. Furthermore, Pin1 promotes several hallmarks of cancer through inactivating 26 tumor suppressors and activating 56 oncogenes [132,133]. By catalyzing the cis/trans conformational change of the target protein, such as Myc, isomerases like Pin1 gain control of the target protein’s stability, activity, and localization [134]. As mentioned, Myc’s stability is regulated through phosphorylation on S62 and T58 and these sites are recognized by *trans*-specific phosphatases; therefore, Pin1 can stabilize Myc in the *cis*-confirmation and prevent degradation [135]. On the contrary, Pin1 can revert Myc back to the *trans*-confirmation after phosphorylation of T58, which allows PP2A to remove phosphorylation from S62 to promote Myc degradation [28]. However, another consideration is that PP2A is commonly inactivated in cancers (described below), and so even if Pin1 reverts Myc back to the *trans*-confirmation, Myc would unlikely get degraded in the absence of PP2A activity. More research is needed to better understand how T58 phosphorylation affects Pin1 activity and S62 dephosphorylation. Additionally, Pin1 can promote self-ubiquitination of Fbw7, the E3 ubiquitin ligase that ultimately degrades Myc [136]. Pin1’s influence on Myc’s transcriptional activity and stability potentiates tumorigenesis and is a potential therapeutic target for *MYC*-overexpressing cells [132,135,136,137].

Lastly, PP2A is a ubiquitously expressed tumor suppressor that accounts for a majority of the phosphatase activity in cells and dephosphorylates a range of substrates such as Akt, p53, β-catenin, and Myc [138]. The holoenzyme can contain a variety of different scaffold (A) and regulatory (B) subunits with a common catalytic (C) subunit, with multiple isoforms for each subunit [139]. Inactivation of PP2A through PP2A inhibitor okadaic acid results in tumorigenesis and cellular transformation [140]. PP2A is commonly inactivated in cancer, including lung, colon, breast, skin, cervix, and ovarian [139]. This PP2A inactivation occurs through phosphorylation, somatic mutation, or increased expression of endogenous inhibitors such as SET and CIP2A [141,142,143,144]. In the case of Myc, PP2A inactivation prevents dephosphorylation of S62, therefore stabilizing Myc and promoting transformation [28,121]. In sum, inhibition of GSK3 through PI3K, overexpression of Pin1, and inactivation of PP2A promote stability of Myc (Figure 5). Although there are many opportunities to increase Myc stability, many of these proteins are also potential therapeutic targets to promote Myc’s degradation.

#### Enhancing Degradation of Myc

Given the various levels regulating Myc degradation, numerous compounds have been developed to enhance Myc degradation through inhibition of PI3K or Pin1 and re-activation of PP2A. Becker and collogues demonstrated efficacy in combining a PI3K inhibitor with a microtubule destabilizer in high-Myc expressing cells. First, they eloquently demonstrated that unphosphorylated S62-Myc binds to mitotic tubules and is protected from degradation [145]. Given this interaction, treatment with a microtubule destabilizer, vincristine, drastically reduced Myc protein and P493-6 B-cell lymphoma cells with ectopic Myc expression were more sensitive to colony forming unit inhibition than Myc low-expressing cell lines. Since PI3K/AKT inhibits GSK3B activity and therefore stabilizes Myc, Becker and collogues investigated the addition of PI3K inhibitor idelalisib following the G2-M arrest induced by vincristine. Treating first with vincristine followed by idelalisib led to higher cell death and decreased clonogenic growth than either compound alone across 16 Burkitt lymphoma and DLBCL cell lines [145]. Furthermore, this combination lead to reduction of Myc and tumor viability in two lymphoma in vivo models, in which the compounds as single agents were not effective. These results suggest a novel avenue of disrupting Myc stability via microtubule destabilizers followed by PI3K inhibition to further decrease Myc protein levels. Another targetable signaling pathway that influences Myc degradation is the MEK/ERK pathway. As mentioned, ERK maintains S62 phosphorylation of Myc, which promotes Myc’s stability [120]. Therefore, inhibition of the MEK/ERK pathway through MEK inhibitor U0126 reduced Myc expression and growth in rhabdomyosarcoma cell lines [146]. Furthermore, inhibition of the MEK/ERK pathway or the consequent decrease in Myc expression, a known driver of radioresistance, sensitizes cancer cells to radiation therapy [147,148].

Aside from Pin1′s influence over Myc’s stability, there are several other mechanisms in which Pin1 can promote tumorigenesis such as sustaining proliferative signaling and downregulating tumor suppressors [132]. More than ten Pin1 inhibitors have been developed that demonstrate anticancer activity, including sensitizing various cancer cells to chemotherapy [132]. We will discuss two Pin1 inhibitors—All-trans retinoic acid (ATRA) and KPT-6566, that have more favorable specificity and safety profiles than other Pin1 inhibitors. ATRA is clinically used for acute promyelocytic leukemia (APL), although its drug target was unknown. Through a mechanism-based high throughput screen, Wei and collogues discovered ATRA directly binds and degrades Pin1 [149]. ATRA was capable of decreasing Pin1 and tumor growth in APL mouse models and APL human patients’ bone marrow, along with in vivo models of triple negative breast cancer [149] and acute myeloid leukemia [150]; both cancers overexpress Pin1. However, ATRA has a short half-life of 45 minutes and moderate anti-cancer activity. Yang and collogues developed an improved, controlled-release formulation of ATRA (ATRA-PLLA microparticles) that demonstrated selectivity for Pin1 inhibition and improved anti-cancer efficacy in xenografts of hepatocellular carcinoma, a cancer that is enhanced by Pin1 [151]. Several other liposomal ATRA delivery methods have been developed and performed well in clinical trials for APL patients [152], although it appears trials for solid tumors utilizing the improved ATRA formulation are lacking. Additionally, these studies did not specifically investigate the effects of ATRA and Myc. Several older studies across small cell lung cancer, breast cancer, and colon cancer demonstrated treatment with ATRA decreased Myc expression at the gene or protein level [153,154,155]. Selective Pin1 inhibitor KPT-6566, which was also identified through a mechanism-based screen, sets Pin1 for degradation. When KPT-6566 binds to the catalytic site of Pin1, reactive oxygen species are produced and DNA damage occurs, leading to cell death particularly in *Pin1*-overexpressing cancer cells [156]. There are no data on KPT-6566 decreasing tumor volume in vivo, but in mice injected with MDA-MB-231 cells, KPT-6566 daily treatment reduced metastatic spread and showed no toxicities in vital organs [156]. Again, these studies did not investigate the effects of Pin1 inhibition on Myc. More development is necessary to improve efficacy and drug-likeness of Pin1 inhibitors, especially in the context of Myc-driven cancers.

Compounds that target PP2A, which is the main phosphatase the regulates Myc stability, have shown promise in promoting Myc degradation and cell death. There are several methods published on indirectly activating PP2A as an anti-cancer treatment, such as antagonizing the endogenous PP2A inhibitors SET (via OP449 [41,157] or FTY720 [158,159]), and CIP2A (via bortezomib, erlotinib, or celastrol) or disrupting PP2A post translational modifications [143]. SET-inhibitor OP449 increased PP2A activity dose-dependently and OP449-treated leukemia xenografts had a two-fold reduction of tumor burden [157]. In breast cancer, OP449 decreased both phosphorylation levels of S62-Myc and Myc transcriptional activity across several cell lines in vitro. OP449 additionally induced apoptosis while reducing tumor volume and increasing PP2A activity in vivo [41]. In terms of disrupting CIP2A, the described inhibitors were primarily discovered as a proteasome inhibitor (bortezomib), EGFR kinase inhibitor (erlotinib), or anti-cancer (celastrol), but indirectly or independently reduce CIP2A expression or activity [143,160]. Small molecule activators of PP2A (SMAPs) have also emerged as a new class of validated compounds that re-activate PP2A through binding to the A scaffolding subunit of PP2A. As PP2A reactivates, S62-Myc becomes dephosphorylated and Myc is sent for degradation. Recently, SMAPs demonstrated efficacy through binding to PP2A in in vivo models of Burkitt’s lymphoma, non-small cell lung cancer, and triple-negative breast cancer—all Myc-driven cancers, representing Myc amplification, post-translational stabilization, and overexpression [161]. SMAPs also display efficacy in prostate and pancreatic cancer models [162,163]. Dr. Narla, one of the developer of SMAPs, serves as Chief Scientific Officer for Rappta Therapeutics to further develop these anti-cancer molecules that reactive PP2A [164,165]. In all, targeting Myc’s protein stability may help reduce toxicity that is expected with a complete loss of Myc.

## 4. Taking Advantage of *MYC* Overexpression to Initiate Synthetic Dosage Lethality in the Context of Cell Cycle

Since transcription factors pose as difficult drug targets, leveraging synthetic lethality offers an alternative approach of antitumoral therapy. Synthetic lethality occurs when a mutation or inhibition of two specific genes leads to cell death, but a mutation or inhibition of just one gene does not affect viability [166]. Synthetic *dosage* lethality is when manipulation of expression levels leads to cell death; for example, overexpression of gene A and presence of gene B is viable, but the combination of gene A overexpression and loss or lower expression of gene B results in cell death. Therefore, synthetic lethality, or more specifically synthetic dosage lethality, can be advantageous in cancer as the tumors already have mutations or oncogenic addiction, such as overexpression of *MYC*. A synthetic lethal approach affects the mutated tumor cells and spares the normal cells.

Synthetic lethal targets are identified in an unbiased, high-throughput fashion through RNA interference (RNAi) or CRISPR screens on isogenic cells—cells that differ by a mutation in a single gene. Although the idea sounds swift, identifying clinically relevant synthetic lethal interactions have proven difficult due to validation of lethal mutants by recovery, condition-dependent interactions, and rarity [166]. However, PARP inhibitors successfully demonstrated this concept clinically when given to cancer patients with *BRCA* mutations, such as in breast and ovarian cancer [167,168].

Understanding the biological results of *MYC* overexpression will help identify second-site targets that lead to synthetic lethality. Reports show that *MYC*-overexpressing cancer cells have increased sensitivity to apoptosis in response to cytotoxic drugs or radiation [169]. However, the opposite appears to be true in melanoma, in which lower *MYC* expression improves susceptibility to chemotherapy and radiation due to reactive oxygen species production and mismatch repair protein inhibition [170,171]. As Myc is a master regulator of cell proliferation and metabolism, genes affiliated with these processes offer a promising avenue to identify synthetic lethal targets.

Cells overexpressing *MYC* have more mitotic abnormalities, such as altered spindle morphology and mitotic timing [172]. During mitotic stress, Myc worsens mitotic dysfunction and enhances apoptosis, which explains the many cell cycle proteins as targets for synthetic lethality. In normal conditions, advancing through the cell cycle phases of G1, S, G2, and M requires four heterodimers of cyclin-dependent serine/threonine kinases (CDK) and cyclins: CDK1, 2, 4, 6, and cyclins A, B, E, D, all of which are Myc target genes [173]. Cyclin B1 binds to CDK1 at the G2-M transition, activating the complex to promote mitosis. CDK1 is the only essential CDK required for cell cycle progression and it is rarely dysregulated in cancer [174,175]. Inhibiting CDK1 typically results in a G2 arrest, but in *MYC*-overexpressing cells, CDK1 inhibition leads to apoptosis [176,177]. Cyclins and CDKs just scratch the surface of proteins involved in the cell cycle.

Myc also induces expression of Aurora A kinase, which reciprocally stabilizes Myc in addition to its role in cell cycle [178]. Aurora kinases A and B direct cell cycle progression through G2-M. Aurora A aids in centrosome function, spindle assembly, and mitotic entry while Aurora kinase B is the catalytic component within the chromosomal passenger protein complex (CPPC) to control chromosomal condensation and cytokinesis [179,180]. Aurora A and B kinases are overexpressed in breast and colon cancers, along with sarcoma, esophageal, and stomach cancers [181]. Myc is known to upregulate Aurora kinase A and B expression in B-cell lymphomas, which is necessary to maintain the lymphoma [182]. Similarly, overexpressing *MYC* in medulloblastoma cell lines with low *MYC* expression led to an associated increase in Aurora B expression [183]. Overall, cell cycle proteins such as Aurora kinases or CDK1 in *MYC*-overexpressing cells are potential therapeutic targets as inhibition leads to synthetic dosage lethality (Figure 6).

### Targeting Cell Cycle Proteins

*CDK1:* Purvalanol is a potent CDK1 inhibitor that has selectivity for CDK1 over CDK2 at a low concentration of 4 nM [184]. Goga et al. discovered inhibiting CDK1 pharmacologically with purvalanol or genetically using a cell line with temperature-sensitive *Cdk1* allele results in apoptosis in *MYC*-overexpressing cells [176]. This synthetic lethal interaction led to decreased tumor growth in *MYC*-expressing lymphoma and hepatoblastoma in vivo models. However, purvalanol is not suitable clinically as it is poorly soluble; new variations are needed to pursue CDK1 inhibitors as a clinical candidate. Additionally, Goga et al. also demonstrated promise for targeting survivin, an endogenous inhibitor of apoptosis and known CDK1 target. CDK1 inhibition via purvalnol degraded survivin, and depleting survivin independently resulted in similar results to CDK1 inhibition in that *MYC*-overexpressing cells were more sensitive to survivin degradation via peptide inhibitors [176]. Given that loss of p53 influences decreased apoptosis in *MYC*-overexpressing cells [185,186], the authors explored effects of p53 status on the efficacy of the CDK1 inhibitor. Through the use of wildtype and p53^−/−^
*MYC* overexpressing mouse embryonic fibroblasts, Goga et al. determined p53 status is independent of purvalanol-induced apoptosis [176]. This is advantageous as many cancers have p53 deficiencies.

A second study from the same group reported that triple-negative breast cancer (TNBC) with elevated *MYC* expression displayed efficacy with CDK1 inhibition (purvalanol, dinaciclib, or siRNA), compared to lines with low *MYC* expression [187]. Although third-generation CDK inhibitor dinaciclib inhibits CDK1, CDK2, CDK5, and CDK9, efficacy in TNBC cell lines was shown to be specific to CDK1 inhibition by knocking down *Cdk1* via siRNA. Furthermore, dinaciclib, which has improved pharmacokinetic and pharmacodynamic properties than previous CDK inhibitors, decreased tumor volume by about 50% in TNBC xenograft mice [187]. Several clinical trials testing dinaciclib in mainly hematologic malignancies recently completed and are pending results. In 2015 studies, 11% of relapsed multiple myeloma patients partially responded and 54% of lymphocytic leukemia patients partially responded [188,189]. In a randomized phase II in TNBC patients, dinaciclib failed to outperform capecitabine, the standard of care [190]. These studies did not consider *MYC* expression but offer promising results to continue dinaciclib in clinical research. An active Phase I clinical trial, NCT01676753, is assessing *MYC* overexpression with dinaciclib + pembrolizumab efficacy in advanced breast cancer [191].

*Aurora Kinases*: Given the similarities and genetic overlap between Aurora kinases A and B, many aurora kinase inhibitors are nonselective, other than Alisertib (MLN8237), which is specific to Aurora A and Barasertib (AZD1152), which is specific to Aurora B. VX-680 and AZD1152 have been studied specifically in *MYC*-overexpressing cancers preclinically. Yang et al. demonstrated proof of concept of synthetic lethality with non-specific aurora kinase inhibitor VX-680 [192]. Pulse treatment of VX-680 in Myc-driven models of lymphoma resulted in a 3-fold increase in survival. Yang and collogues propose synthetic lethality is a result of failed spindle checkpoint due to inhibition of Aurora B (resulting in a compromised CPPC) and *MYC* overexpression leads to polyploidy; the combination of the two proceeds to apoptosis. Yang and Goga both acknowledged defects in CPPC, through either survivin or Aurora B inhibition, which led to synthetic lethality in *MYC*-overexpressing cells. This suggests inhibition of other CPPC components as additional avenues to explore [176,192]. Furthermore, similar to Goga et al.’s findings, Yang et al. reported that the synthetic lethal interaction between Aurora kinase inhibitor and Myc is also independent of p53.

Lastly, another p53-deficient cancer, small cell lung cancer (SCLC), is also susceptible to Aurora kinase-Myc synthetic lethality. Helfirch et al. demonstrated *MYC* amplification is a good biomarker for predicting in vitro and in vivo growth inhibition of SCLC upon AZD1152 treatment [193]. Similarly, *MYC*-overexpressing medulloblastoma cells treated with AZD1152 were more sensitive to apoptosis than the low-expressing parent cell line, as was true for medulloblastoma cells with endogenous *MYC* overexpression [183]. Medulloblastoma xenograft models treated with AZD1152 had decreased tumor growth and prolonged survival.

Several Aurora kinase inhibitors have been tested clinically, including MK-0457 (or VX-680), AZD1152, PHA-739358, and MLN8237 [180]. Clinical trials that included *MYC* expression as a biomarker appear to be limited to Aurora A inhibitor Alisertib [194,195]. Recent in vivo studies of Alisterib in *MYC*-overexpressing lymphoma xenografts demonstrated synthetic lethality by caspase-independent cell death and complete tumor regression when paired with chemotherapy cyclophosphamide [196].

In vivo reports of CDK1 and Aurora kinase inhibitors support the inclusion of *MYC* overexpression as a recruitment factor in clinical trials or at least warrant further investigation of *MYC* a biomarker for these inhibitors. This stratification may result in an improved clinical outcome. Additional potential synthetic lethal interactions in Myc-driven cancers are described by Cermelli et al. [197].

## 5. Myc Drives Metabolism through Its Target Genes

Cellular proliferation is closely related to metabolism. In cancer, metabolic reprogramming, such as prompt ATP synthesis, increased anabolism of macromolecules, and redox homeostasis, support the rapidly proliferating cancer cells [198]. Understandably, Myc, the master regulator of growth, also aids in metabolic reprogramming. Several Myc target genes are involved in metabolic pathways, including glucose transporter GLUT1 (*SLC2A1*), glutaminase (*GLS*), hexokinase 2 (*HK2*), phosphofructokinase (*PFKM*), enolase 1 (*ENO1*), peroxisome proliferator-activated receptor gamma coactivator 1-beta (*PGC-1β*), nuclear respiratory factor 1 (*NRF1*), and inosine monophosphate dehydrogenase (*IMPDH1/2*) [199,200]. Through Myc’s many target genes, it can regulate aerobic glycolysis (the Warburg effect), mitochondria, and ribosome biogenesis, and metabolism of nucleotides, amino acids, and lipids, which contributes to its oncogenic function [198,199].

Cancer cells have increased levels of guanosine triphosphate (GTP), an energy source and signaling molecule [201]. Both GTP and the rate-limiting enzyme for GTP synthesis, inosine monophosphate dehydrogenase (IMPDH), are Myc target genes [202]. *MYC* and *IMPDH* expression significantly correlated and *IMPDH* overexpression has been observed in several cancers, including glioblastoma, leukemia, colorectal cancer, and small cell lung cancer [203,204,205,206,207]. Manipulating *IMPDH* expression in glioblastoma cells results in the same change in *MYC* expression [207]. Furthermore, Myc activates GTP synthesis and it has been shown in small cell lung cancer (SCLC), that IMPDH is depended upon by naïve and chemoresistant high-*MYC* SCLC cells [208]. In addition, IMPDH links Myc’s role in nucleotide biosynthesis and ribosome biogenesis as IMPDH-dependent GTP synthesis is needed for Pol I synthesis of pre-ribosomal RNA [208].

Glutaminase, another target gene of Myc, converts the abundant glutamine into glutamate. Myc-driven cancers depend on glutamine metabolism rather than glucose, especially when deprived of oxygen [209,210,211]. Furthermore, glutamine-depletion-induced apoptosis is dependent on Myc activity [212], and on the other hand, Myc-induced renal adenocarcinoma depends on glutaminase [213]. Shen et al. explored the concept of glutaminase dependency in the context of ovarian cancer, in which >45% of patients’ tumors overexpress *MYC*. Elevated *MYC* expression correlated with glutaminase in immortalized cell lines and primary cultures, and overexpression of *MYC* and *GLS* were associated with chemoresistance and worse disease outcome [214].

We will discuss inhibiting the proteins of metabolic Myc target genes, IMPDH and glutaminase, along with epigenetic cofactor WDR5 that recruits Myc to drive ribosome biogenesis (Figure 7). Details of Myc’s role in cancer metabolism are beyond the scope of this review and are summarized in several articles [17,198,199,215]. Additional potential Myc-driven metabolic targets are described by Dong et al. [198].

### Targeting Metabolism Through Myc Target Genes and Cofactors

Myc’s effects on metabolism are highlighted by the negative impact of various metabolic inhibitors in Myc-driven cancer models. First, there are two clinically available IMPDH inhibitors: mycophenolic acid (MPA) and mizoribine. Both are clinically used as an immunosuppressants to prevent organ transplant rejection, but additional research is necessary to explore their anticancer properties. MPA appears to primarily serve as an anticancer tool; it has been preclinically tested in several cancers, but has dose-limiting toxicity due to gastro-intestinal side effects [216,217,218]. However, mizoribine has higher tolerability [219]. Studies connected the anticancer efficacy of IMPDH inhibitors to Myc; it appears that Myc is needed for the antiangiogenic properties of MPA [218]. Huang et al. show Myc-driven SCLC and hepatoblastoma demonstrate sensitivity to mizoribine and MPA, confirming a dependence on IMPDH. Importantly, mizoribine was capable of decreasing tumor growth in immunocompetent mice, despite its immunosuppressive properties [208]. This study warrants further testing and development of these clinically-available IMPDH inhibitors in Myc-driven cancer models.

A second approach is targeting glutaminase in *MYC*-overexpressing cells. The concept of pharmacologically inhibiting glutaminase has been discussed since 1975 [220], but there were concerns of targeting a major metabolic component. Allosteric glutaminase inhibitor Bis-2-(5-phenylacetamido-1,2,4-thiadiazol-2-yl)ethyl sulfide (BPTES) has been studied extensively. Treatment with BPTES increases reactive oxygen species production and hinders cell bioenergetics, leading to cell death [211]. In an in vivo renal adenocarcinoma model, BPTES reduced tumor growth by 32% [213]. However, modifications must be made to BPTES to improve its therapeutic potential as it has a moderate potency and poor solubility [221]. The newest glutaminase inhibitor, CB-839, is a BPTES derivative and is currently in Phase II clinical trials for several cancers including colorectal cancer, acute myeloid leukemia, and triple negative breast cancer. Shen et al. demonstrated preclinically that ovarian tumor xenografts treated with CB-839 resulted in increased sensitivity to PARP inhibitor olaparib as glutaminase inhibition led to replicative stress [214]. A Phase Ib/II study investigating CB-839 in combination with PARP inhibitor talazoparib for solid tumors is currently recruiting patients (NCT03875313) and a Phase I study combining CB-839 with PARP inhibitor niraparib in platinum resistance BRCA-wildtype ovarian cancer has posted (NCT03944902).

As mentioned, Myc’s role in metabolism primarily stems from expression of its target genes that are involved in metabolic pathways. The previously described Myc metabolic therapeutics inhibit Myc’s target genes, but not Myc’s activity. Thomas et al. discovered that epigenetic cofactor WDR5 recruits Myc to chromatin to promote expression of genes involved in biomass accumulation [222]. An inducible exon swap system in a Burkitt lymphoma cell line was created to study the interaction between Myc and WDR5 by implementing a mutant Myc that could not interact with WDR5. Inhibition of WDR5 prevented Myc’s function as a transcription factor by disrupting gene binding, which decreased transcription of translational machinery, including ribosome protein subunits and nucleolar RNAs. When the exon swap system was assessed in vivo, switching to the WDR5-interaction-defective Myc resulted in apoptosis, decreased tumor volume, and improved survival [222]. Thomas and collogues additionally reported that targeting the “WIN” site in WDR5 may also be a valuable target to displace Myc from chromatin. WDR5 inhibitors are currently being synthesized to further study the anticancer effects of disrupting the Myc-WDR5 interaction [223]. Targeting an epigenetic cofactor that aids in Myc’s activity will prevent Myc’s target gene from being transcribed; this route may be more beneficial than inhibiting already-transcribed genes under Myc’s control. This study opens an avenue outside the context of metabolism to explore other targetable Myc-interacting cofactors to prevent Myc binding to chromatin.

## 6. Conclusions

This review summarizes the main mechanisms by which c-Myc promotes tumorigenesis and the different therapeutic approaches that directly/indirectly target Myc. Broadly, we described inhibitors that prevent Myc’s actions as a transcription factor through altering Myc stability (transcription, dimerization, degradation), inducing synthetic lethality via cell cycle targets, and inhibiting Myc target genes involved in metabolism. There are other methods of disrupting Myc’s activity not listed here, including inhibiting Myc-recruited cofactors or epigenetic mechanisms [13,25,122]. Additionally, Myc’s robust control on microRNA expression is another area of interest [224,225,226], given Myc’s ability to activate oncogenic miRNAs and repress tumor suppressive miRNAs [225,227]. Modulating expression of the miRNAs to promote anticancer effects as a therapeutic option is being explored [228,229]. When clinically testing these therapeutics in Myc-driven cancers, it is important to consider patients’ *MYC* expression in the case of stratifying patients and identifying clinically relevant subgroup results. Furthermore, although this review was limited to well-studied c-Myc, the Myc family of c-Myc, N-Myc and L-Myc, can be functionally redundant, and therefore inhibition of the Myc family rather than one specific Myc may be required [230]. All in all, new discoveries improved our understanding of the “myc-anisms” behind Myc-driven cancers and enhanced the potential of targeting the “undruggable” Myc.

## Figures and Tables

**Figure 1 ijms-21-09486-f001:**
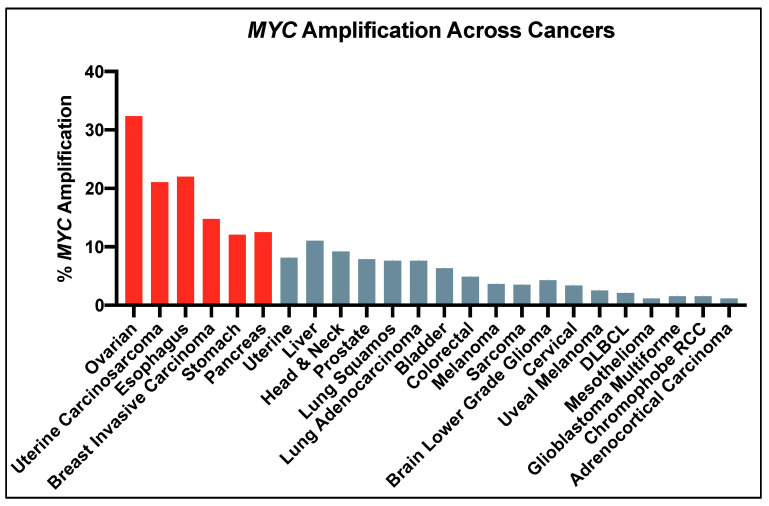
*MYC* amplification across cancers. Percentage represents number of patients with *MYC* amplification for that cancer type. Red bars represent cancers in which >10% of patients harbor *MYC* mutations. Data from The Cancer Genome Atlas Pan Cancer 2018 Dataset, cancer.gov/TCGA.

**Figure 2 ijms-21-09486-f002:**
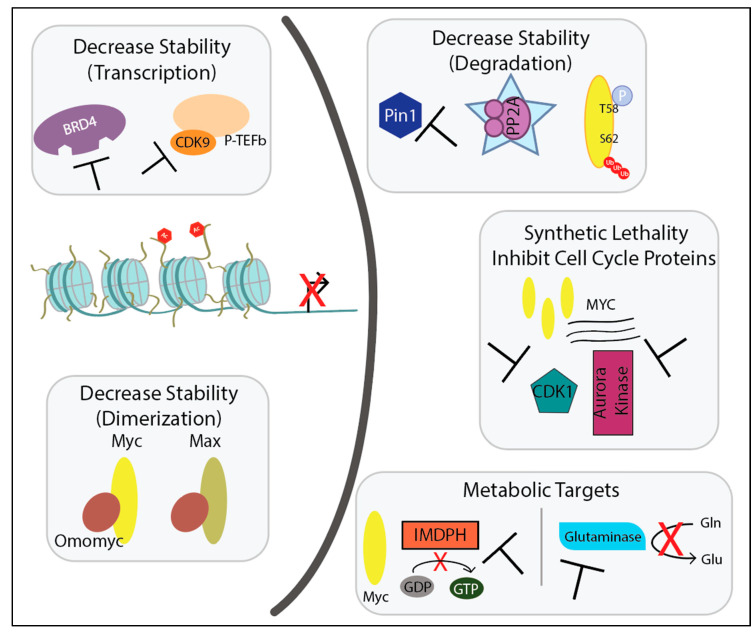
Schematic presenting the various cellular processes to target through inhibition or reactivation in the nucleus (**left**) or cytoplasm (**right**) upon Myc-induced tumorigenesis. All will be described in detail in this review.

**Figure 3 ijms-21-09486-f003:**
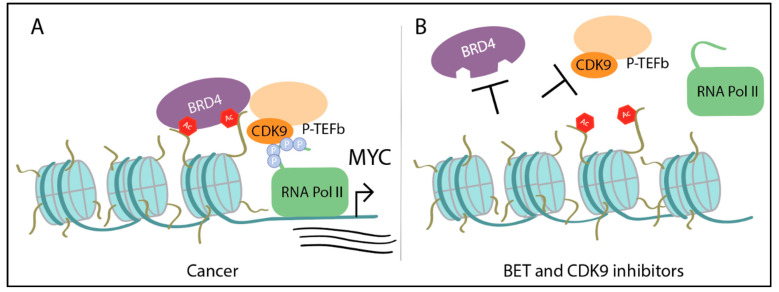
Upon *MYC* amplification in cancer, Myc recruits additional transcriptional cofactors to drive transcription: (**A**) BRD4 binds to acetylated lysines on histone tails and recruits P-TEFb (which includes CDK9), that phosphorylates the carboxy terminal domain of RNA Pol II. Myc can also individually recruit P-TEFb. (**B**) Treatment with bromodomain and extra-terminal motif (BET)/BRD4 inhibitors prevents BRD4 from binding to histone tails and treatment with CDK9 inhibitors disrupts CDK9′s kinase activity. Thus, both result in failure of activating transcription of *MYC* or Myc target genes.

**Figure 4 ijms-21-09486-f004:**
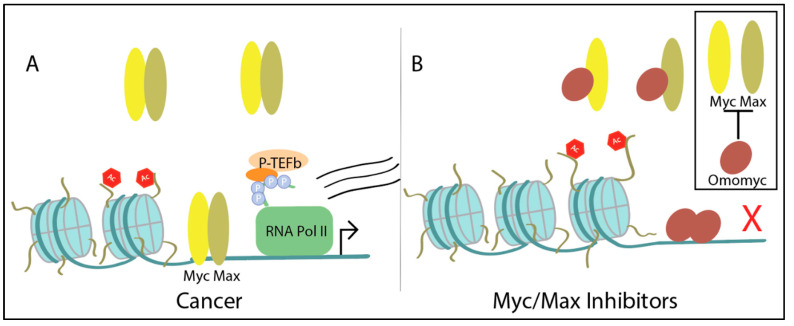
Heterodimerization with Max is required for Myc’s oncogenic activity: (**A**) Upon heterodimerization, Myc/Max binds to the E-box and initiates transcription. This is a normal cellular process, but in cancer, Myc amplification further increases Myc activity; (**B**) Treating with Omomyc, a dimerization inhibitor that preferentially binds to Max or homodimerizes, displaces Myc at E-boxes and decreases Myc transcription. Other discussed Myc/Max inhibitors either disrupt Myc/Max dimers or block Myc’s interaction with DNA, but not both.

**Figure 5 ijms-21-09486-f005:**
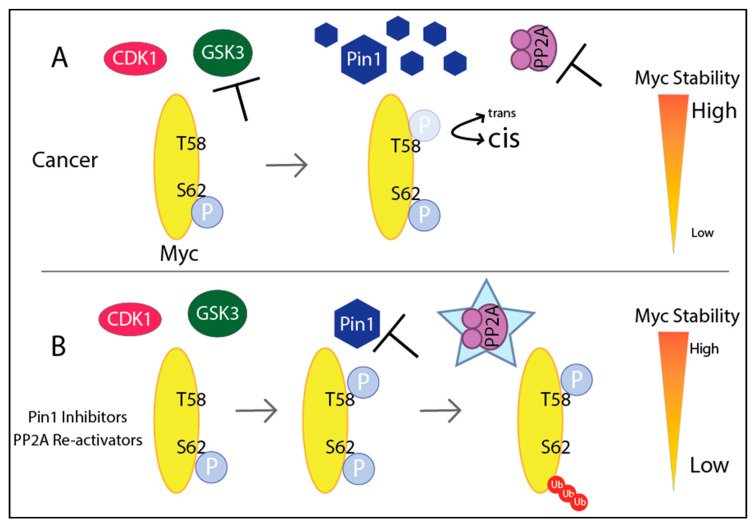
Disrupting Myc stability: (**A**) In cancer, PI3K signaling inactivates GSK3, preventing phosphorylation of T58 Myc. Pin1 overexpression keeps Myc in the *cis*-confirmation, preventing PP2A *trans*-specific enzyme from binding to Myc. Furthermore, PP2A is inactivated in several cancers, and therefore S62 remains phosphorylated. All of this leads to high Myc stability. (**B**) Inhibition of PI3K allows for GSK3 to phosphorylate T58 on Myc, which is required for degradation. Pin1 inhibitors and PP2A activators allow for PP2A to recognize and remove the phosphorylation of S62, leading to low stability and Myc’s degradation.

**Figure 6 ijms-21-09486-f006:**
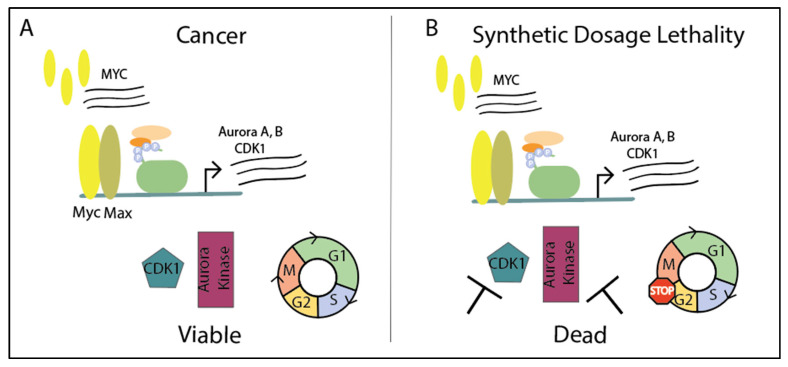
Taking advantage of oncogenic addiction in cancer with synthetic dosage lethality. (**A**) Myc produces target genes Aurora kinases A and B and CDK1, which help stabilize Myc and promote the cell cycle. *MYC* is commonly amplified in cancer and cells remain viable. (**B**) In *MYC*-overexpressing cancers, inhibiting CDK1 or Aurora kinases leads to cell cycle arrest and synthetic dosage lethality.

**Figure 7 ijms-21-09486-f007:**
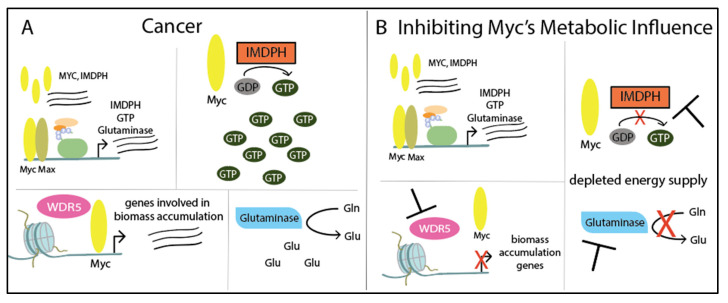
Myc influences metabolism through its target genes: (**A**) In cancer, *MYC* overexpression correlates with *IMPDH* expression and transcribes other target genes, including *GTP* and glutaminase (*GLN*). IMPDH catalyzes GDP to GTP and glutaminase converts glutamine (Gln) to glutamate (Glu), a major energy source in cancer. Lastly, epigenetic co-factor WDR5 recruits Myc to chromatin to express genes involved in biomass accumulation. (**B**) Although Myc still has control over target genes *IMPDH*, *GTP*, and *GLN*, the function of these proteins can be inhibited. Cancer’s energy supply can be depleted by inhibiting IMPDH, which prevents GTP production, or by inhibiting glutaminase, which will limit the pool of glutamate. Finally, inhibiting WDR5 will prevent Myc’s target gene expression of biomass related genes.

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
