# Peer review of "The Molecular ‘Myc-anisms’ behind Myc-Driven Tumorigenesis and the Relevant Myc-Directed Therapeutics"

_ijms, 2020, doi:10.3390/ijms21249486_

Round 1

Reviewer 1 Report

Dear Authors Jessica McAnulty and Analisa DiFeo,

first of all I wanted to congratulate you on the work done. c-Myc is probably one of the oldest and most studied oncogenes yet, it is still extremely relevant today. Your manuscript analyzes in detail every mechanism, from the genetic to the post-translational one potentially responsible for the aberrant accumulation / function of c-Myc. In this way you provide a panel of possible strategies for targeting c-Myc for therapeutic purposes.
I found the complete review. I only allow myself to underline how some references, which I believe are important and worth mentioning, are missing.

For example, when discussing the role of MAPK kinases in post-translational regulation of the c-Myc protein, the authors should include the work of one of the research groups that was among the first to demonstrate this mechanism (Down-regulation of c-Myc following MEK/ERK inhibition halts the expression of malignant phenotype in rhabdomyosarcoma and in non muscle-derived human tumors. Mol Cancer. 2006 Aug 9;5:31. doi: 10.1186/1476-4598-5-31).

The authors mention the role of c-Myc in the phenomena of chemoresistance but forget to mention some works that demonstrate the role of this oncoprotein in radio resistance. Radiation therapy is as much an indispensable therapy tool as chemotherapy. Radioresistance, also largely dependent on c-Myc, is a subject of indisputable importance (c-Myc Sustains Transformed Phenotype and Promotes Radioresistance of Embryonal Rhabdomyosarcoma Cell Lines. Radiat Res. 2016 Apr;185(4):411-22. doi: 10.1667/RR14237.1. andDisruption of MEK/ERK/c-Myc signaling radiosensitizes prostate cancer cells in vitro and in vivo. J Cancer Res Clin Oncol. 2018 Sep;144(9):1685-1699. doi: 10.1007/s00432-018-2696-3. Epub 2018 Jun 29.).

Author Response

Suggestions from Reviewer 1:

  1. The three suggested papers were included in Section 3.3.1 Enhancing degradation of Myc (page 11 of 36).

Reviewer 2 Report

In the review entitled “The molecular ‘Myc-anisms’ behind Myc-driven tumorigenesis and the relevant Myc-directed therapeutics” the authors explores the various mechanisms in which Myc promotes cancer and highlights key therapeutic approaches to disrupt Myc activity. In particular, they focuses on how Myc’s oncogenic activation leads to tumorigenesis through initiating transcription, increasing stability, and influencing the cell cycle and metabolism, coupled with descriptions of the Myc inhibitors that target each mechanism.

The review is clear and almost always simple to read and it is focused both on general concepts about the involvement of this factor in cancer and in possible therapeutic treatments to counteract its function.

I think there are only some aspects that the authors should revise or better elaborate, as reported below:

Figure 1 can be refined to be more comprehensive and the authors should revise it to clarify the goal of the review and the ways by which they aim at reaching the result.

Do you have any kind of evidence about synthetic dosage lethality in the context of cell cycle with a particular emphasis to the relationship between Myc and p53?

Is it there also any evidence of correlation with microRNA network or about a way to act upon MYCN by regulating microRNAs expression?

At page 11 of 28, lane 392, it is reported the sentence: “Another Pin1-overexpressing cancer…” and the it is truncated. It lacks the end of the sentence.

In general, the entire article text need to be slighlty revised for its English grammatical style and form. There are some terms mistake along the paper.

In conclusion, I think that the review can be considered for publication after some small revisions or integrations regarding the points reported above.

Author Response

Suggestions from Reviewer 2:

  1. Figure 1 (page 4 of 36) was re-designed to summarize the various types of inhibitors described in this review.
  2. The influence of p53 status in Myc overexpressing cancers in the context of synthetic lethality was included on pages 13 and 14 of 36.
  3. A description of Myc’s influence on microRNAs was incorporated in Section 6 Conclusion, page 16 of 36.
  4. The misplaced, truncated sentence on page 11 of 36, line 392 was removed.
  5. The manuscript was reviewed in its entirety by both authors to identify and revise grammatical style and form.